# The Efficacy of Hybrid Vaginal Ovules for Co-Delivery of Curcumin and Miconazole against *Candida albicans*

**DOI:** 10.3390/pharmaceutics16030312

**Published:** 2024-02-23

**Authors:** Brenda Maria Silva Bezerra, Sara Efigênia Dantas de Mendonça y Araújo, José de Oliveira Alves-Júnior, Bolívar Ponciano Goulart de Lima Damasceno, João Augusto Oshiro-Junior

**Affiliations:** 1Pharmaceutical Sciences Postgraduate Program, Center for Biological and Health Sciences, State University of Paraíba, Av. Juvêncio Arruda, S/N, Campina Grande 58429-500, PB, Brazil; brendamariasb@outlook.com (B.M.S.B.); saraedm1999@gmail.com (S.E.D.d.M.y.A.); bolivarpgld@servidor.uepb.edu.br (B.P.G.d.L.D.); 2Department of Pharmacy, Center for Biological and Health Sciences, State University of Paraíba, Av. Juvêncio Arruda, S/N, Campina Grande 58429-500, PB, Brazil; oliveiraj.alves@yahoo.com

**Keywords:** vulvovaginal candidiasis, ureasil–polyether, sol-gel process, combination antifungal therapy

## Abstract

Curcumin (CUR) is a natural compound that can be combined with miconazole (MCZ) to improve vulvovaginal candidiasis (VVC) caused by *Candida albicans* treatment’s efficacy. This study aimed to develop ureasil–polyether (U-PEO) vaginal ovules loaded with CUR and MCZ for the treatment of VVC. Physicochemical characterization was performed by thermogravimetry (TGA), differential thermal analysis (DTA), Fourier transform infrared spectroscopy (FTIR), and in vitro release. Antifungal assays were used to determine minimum inhibitory concentrations (MICs) and synergism between CUR and MCZ, and the activity of U-PEO ovules were performed by microdilution and agar diffusion. TGA results showed high thermal stability of the hybrid ovules. In DTA, the amorphous character of U-PEO and a possible interaction between CUR and MCZ were observed. FTIR showed no chemical incompatibility between the drugs. In vitro release resulted in 80% of CUR and 95% of MCZ released within 144 h. The MICs of CUR and MCZ were 256 and 2.5 µg/mL, respectively. After combining the drugs, the MIC of MCZ decreased four-fold to 0.625 µg/mL, while that of CUR decreased eight-fold to 32 µg/mL. Synergism was confirmed by the fractional inhibitory concentration index (FICI) equal to 0.375. U-PEO alone showed no antifungal activity. U-PEO/MCZ and U-PEO/CUR/MCZ ovules showed the greatest zones of inhibition (≥18 mm). The results highlight the potential of the ovules to be administered at a lower frequency and at reduced doses compared to available formulations.

## 1. Introduction

*Candida albicans* is an opportunistic fungus that naturally colonizes the vaginal mucosa. However, certain changes in the microenvironment and the microbiota of the region, due to the use of antibiotics, contraceptives, and corticosteroids (among other factors) may cause the fungus to grow pathogenically and lead to an infection known as vulvovaginal candidiasis (VVC) [1,2]. VVC affects 70% of all women of reproductive age at least once in their lifetime and is associated with inflammatory signs and symptoms that cause intense discomfort [3].

The standard treatment for VVC involves the use of azole antifungal agents, either topically or orally. These antifungal agents are preferred due to their moderate safety profile as well as the variety of formulations available [4]. However, administration of these drugs has limitations, as they are contraindicated in pregnancy and are associated with vaginal and gastrointestinal adverse effects that can reduce compliance with treatment. The emerging resistance of *Candida* to azoles is another factor that can trigger therapeutic failure and prompt a search for new treatment options [5,6,7].

In this sense, the combined therapy of synergistic agents seems to be promising. The combination of antifungals is related to increased fungicidal activity, making it possible to reduce doses and adverse effects, as well as offering a lower risk of clinical failure, especially when it comes to strains that are resistant to the currently available drugs [8,9].

Curcumin (CUR) is a polyphenol extracted from the rhizome of *Curcuma longa*, which has been highlighted for its antifungal potential, proving to be 2.5 times more effective than fluconazole in inhibiting *Candida* adhesion to the oral epithelium [10]. In addition, CUR exhibits a synergistic effect with azoles and can be combined with agents such as miconazole to improve the treatment outcome of VVC [11].

Miconazole nitrate (MCZ) is an antifungal drug used for the treatment of VVC. It is available as a 2% cream, as well as a 100 or 200 mg suppository. Topical treatment with this drug lasts from three to seven days, depending on the pharmaceutical form and concentration applied. However, high doses and prolonged treatment with MCZ may be necessary for a complete cure [12].

In this context, combining MCZ with CUR in a vaginal formulation could be beneficial in enhancing its clinical efficacy. However, these drugs belong to class II of the biopharmaceutical classification system (BCS) and present some technological and administration challenges due to their low water solubility, which results in low bioavailability and the need for high frequency of administration [13,14]. 

Therefore, controlled release systems may be promising for reducing the inconveniences of administering CUR and MCZ. These new delivery systems have also been studied to reduce doses and administration frequency, minimize adverse effects, and improve patient compliance [15]. Ureasil–polyether hybrid materials stand out for their mechanical strength, easy manipulation, biocompatibility, and the ability to incorporate high quantities of hydrophilic or lipophilic drugs and control their release profile. This makes them an excellent choice for drug delivery applications where targeted and controlled release are desired [16,17,18].

The high flexibility of these materials makes it possible to develop different pharmaceutical forms from these systems, such as vaginal ovules [18]. These formulations usually contain one or more active compounds that is dispersed or dissolved in a base that melts or dissolves at body temperature and can be inserted into the vagina without irritation, as well as ensuring uniform doses and not requiring a high volume of fluid for dissolution [19,20].

In this study, we developed ureasil–poly(ethylene oxide) (U-PEO) vaginal ovules loaded with CUR and MCZ as a new treatment for vulvovaginal candidiasis caused by *C. albicans*. The results showed the potential of the developed ovules to circumvent the limitations of the administration of the active ingredients and to be used at a lower frequency of administration compared to currently available preparations.

## 2. Materials and Methods

### 2.1. Materials

Curcumin (CUR); Miconazole nitrate (MCZ); O,O′-Bis(2-aminopropyl) polypropylene glycol-block-polyethylene glycol-block-polypropylene glycol (PEO); 3-(Triethoxysilyl)propylisocyanate (IsoTrEOS); and resazurin were obtained from Sigma Aldrich^®^ (São Paulo, Brazil). Sabouraud-dextrose Agar; Yeast Nitrogen Base (YNB); and D-(+)-glucose were obtained from Sigma Aldrich^®^ (São Paulo, Brazil). Sabouraud-Dextrose broth belonged to Kasvi^®^ (São José dos Pinhais, Brazil). Phosphate buffered saline (Dulbecco A) pH 7.3 ± 0.2 belonged to Oxoid Ltd. (Basingstoke, UK). Nystatin was obtained from Pharmácia Geranium (São Paulo, Brazil). Ethanol absolute (purity 99.5%) and dimethyl sulfoxide P.A were of analytical grade (Dinâmica Química Contemporânea Ltda, Indaiatuba, Brazil). HCl at 2 M (purity ≥ 36.5%) was supplied by Qhemis^®^ (São Paulo, Brazil). The surfactants Tween 80 P.S and sodium lauryl sulfate P.A were obtained from Dinâmica Química Contemporânea Ltda (Indaiatuba, Brazil).

### 2.2. Synthesis of Ureasil–Polyether Hybrid Ovules

The hybrid precursor (PEO) was obtained by the sol-gel method through the reaction between the functionalized polymer poly(ethylene oxide), with a molecular mass of 500 g·mol^−1^ and a modified 3-isocinatopropyltriethoxysilane alkoxide (IsOTrEOS) (1:2), kept under reflux in ethanol for 24 h at 70 °C. The solvent was then evaporated under vacuum in a rota-evaporator (IKA RV8, Staufen, Germany) at 80 °C for 15 min [21].

Subsequently, hydrolysis and condensation reactions were carried out to form the ovules using 5 g of precursor, 2000 µL of ethanol, 300 µL of water, and 80 µL of the catalyst agent (HCl 2 M), mixed in this order under agitation at 290 rpm on a magnetic stirrer for 3 min. CUR and MCZ were incorporated in this step at a ratio of 2.7% *w*/*w* of the precursor. The solid powders of the drugs were dissolved in the ethanol/water solution, poured into the precursor mass, and the catalyst was added at the final stage. The hybrid materials formed were transferred to plastic ovule molds and kept in a desiccator to dry at room temperature (±25 °C).

### 2.3. Physico-Chemical Characterization of Hybrid Ovules

#### 2.3.1. Thermogravimetry (TGA) 

The curves of CUR, MCZ, and U-PEO loaded or not loaded with the drugs were obtained on a simultaneous thermal analyzer (Shimadzu^TM^-DTG-60, Shimadzu, Kyoto, Japan) under a nitrogen atmosphere with a 100 mL/min flow rate. An average of 10 mg of each sample were placed in alumina crucibles and heated in a temperature range between 40 and 900 °C, with a heating rate of 10 °C·min^−1^. The TA-60WS Version 2.21 software (Shimadzu, Kyoto, Japan) was used to analyze the thermal events in the TGA based on the derivative curve.

#### 2.3.2. Differential Thermal Analysis (DTA) 

DTA curves were obtained on a simultaneous thermal analyzer (Shimadzu^TM^-DTG-60) under a nitrogen atmosphere with a flow rate of 100 mL/min and a heating rate of 10 °C·min^−1^, in the temperature range between 40 and 400 °C. Approximately 2 mg of the samples were weighed and placed in sealed aluminum crucibles using a manual press. The TA-60WS Version 2.21 software (Shimadzu) was used to analyze the DTA curves.

#### 2.3.3. Fourier Transform Infrared Spectroscopy (FTIR)

FTIR spectra of the samples were obtained on an infrared spectrometer (Spectrum 400-Perkin Elmer, Norwalk, CA, USA) with a wavenumber range between 4000 and 650 cm^−1^ and a resolution of 4 cm^−1^. The data was analyzed using Origin^®^ 8.5 software.

### 2.4. In Vitro Drug Release Assay

The release of CUR was analyzed by immersing the ovules in 900 mL of acetate buffer pH 4.2 plus 15% Tween 80 and 5% absolute ethanol to maintain the sink condition, shaking at 50 rpm at 37 °C ± 2 °C as previously validated by Nicolau Costa et al. [22]. At predetermined times (0.25, 0.5, 0.75, 1, 2, 4, 6, 12, 24, 48, 72, 96, 120, and 144 h), 3 mL aliquots of the medium were collected, filtered through a 0.45 µm pore membrane, and analyzed using a UV-Vis spectrophotometer Shimadzu UV-1900 (Shimadzu, São Paulo, Brazil) at 426 nm. The same volume of collected medium was then replaced.

MCZ release was analyzed by immersing the ovules in 200 mL of acetate buffer pH 4.2 with 5% sodium lauryl sulfate, stirring at 50 rpm at 37 °C ± 2 °C. During the periods specified above, 3 mL aliquots of the medium were collected, filtered through a 0.45 µm pore membrane, and analyzed in an UV-Vis spectrophotometer at 272 nm. The volume of the aliquot removed was returned to the medium.

The release kinetics of CUR and MCZ were evaluated by plotting the data of percentage release versus time, adjusted to different mathematical models such as zero-order, Higuchi, Korsmeyer-Peppas, and Weibull, using SigmaPlot 10.0 software (Systat Software Inc., San Jose, CA, USA).

### 2.5. Antifungal Activity of CUR and MCZ

#### 2.5.1. Fungal Strain and Growth Conditions

*C. albicans* ATCC 10231 was obtained from the American Type Culture Collection (ATCC) and maintained in nitrogen-based yeast broth (YNB, Sigma Aldrich^®^, São Paulo, Brazil) supplemented with 100 mM D-(+)-glucose (Sigma Aldrich^®^, São Paulo, Brazil). The microorganism was then subcultured on Sabouraud-Dextrose Agar (SDA, Sigma Aldrich^®^, São Paulo, Brazil) plates and incubated at 37 °C for 48 h for colony growth.

#### 2.5.2. Determination of the Minimum Inhibitory Concentration (MIC)

The antifungal activity of CUR and MCZ was evaluated by determining the minimum inhibitory concentration (MIC). The inoculum of *C. albicans* was prepared with colonies grown on Sabouraud-Dextrose agar for 24 h at 35 °C ± 2 °C suspended in Sabouraud-Dextrose broth (Kasvi^®^) and standardized in a spectrophotometer at 530 nm to obtain a concentration equivalent to 5 × 10^6^ CFU/mL. For the tests below, the suspension was diluted in broth to reach a final concentration of 2.5 × 10^3^ CFU/mL in the wells.

The MICs of CUR and MCZ were determined using the broth microdilution method according to CLSI (Clinical and Laboratory Standards Institute) standard M27, with adaptations [23]. For this, 100 µL of Sabouraud-Dextrose broth was added to the wells of the 96-well plate. Next, solutions of CUR and MCZ in 10% dimethyl sulfoxide (DMSO) were diluted in a volume of 100 µL in the wells to obtain concentrations of 1024 to 0.03125 µg/mL of CUR and 5 to 0.00015269 µg/mL of MCZ. The medium, growth, and vehicle (DMSO) were controlled. Nystatin at 100 ug/mL represented the positive control. Finally, 5 µL of the fungal suspension was inoculated into all the wells. The microplate was incubated at 35 °C ± 2 °C for 24 h.

After 24 h, the test was revealed by adding 40 µL of 0.01% resazurin (Sigma Aldrich^®^) previously filtered through a 0.22 µm membrane. The change in color from purple to pink after 2 h of incubation showed the growth of *C. albicans* in the wells. The MICs were defined as the lowest concentration of the drug capable of inhibiting growth of the microorganism tested.

#### 2.5.3. Checkerboard Test for Synergism Evaluation

The interaction of CUR with MCZ was evaluated using the broth microdilution checkerboard method, performed on a 96-well plate [24]. To perform the experiment, serial dilutions of the CUR and MCZ stock solutions in 10% DMSO were prepared in Sabouraud-Dextrose broth at concentrations of 256 to 4 µg/mL and 2.5 to 0.0390625 µg/mL, respectively, and inoculated into the wells. The medium, medium with inoculum, and vehicle (DMSO) represented the control groups. Finally, 100 µL of the *C. albicans* suspension was added and the microplate was incubated for 24 h at 35 °C ± 2 °C.

The assay was revealed with 40 µL of 0.01% resazurin after 24 h of incubation. The fractional inhibitory concentration index (FICI) was calculated according to Equation (1):(1)$$FICI={FIC}_{CUR}+{FIC}_{MCZ}=\frac{{MIC}_{CURCOMBINATION}}{{MIC}_{CURALONE}}+\frac{{MIC}_{MCZCOMBINATION}}{{MIC}_{MCZALONE}}$$

FICI values ≤ 0.5 indicate synergism, while FICI = 1 and FICI > 2.0 are considered additive and antagonistic, respectively [24].

### 2.6. Antifungal Activity of U-PEO Ovules 

Two methods were used to evaluate the antifungal activity of U-PEO ovules: broth microdilution and agar diffusion. The preparation of the samples and the experimental procedure are described in the topics below.

#### 2.6.1. Sample Preparation

To evaluate the antifungal activity of U-PEO by microdilution, we first crushed and pulverized a pure U-PEO ovule. The resulting powder was weighed to a mass equivalent to the amount of material containing CUR or MCZ in the concentration required to produce the stock solutions used in the microdilution tests of the isolated actives, described in Section 2.5.2. Subsequently, the properly weighed powders were physically mixed with CUR and MCZ in the quantities required to reproduce the stock solutions for the MIC test. Pure U-PEO and the physical mixtures of U-PEO/CUR and U-PEO/MCZ were solubilized in 10% DMSO.

For the agar diffusion test, the ovules of U-PEO, U-PEO/CUR, U-PEO/MCZ, and U-PEO/CUR/MCZ were sectioned into smaller round pieces measuring 6 × 6 mm.

#### 2.6.2. Broth Microdilution 

The microdilution test was conducted as described in Section 2.5.2. The solution of pure U-PEO in 10% DMSO was serially diluted in the microplate from 1024 to 8 µg/mL, as was the solution of the physical mixture of U-PEO/CUR. The solution of U-PEO/MCZ in the same vehicle was diluted in the microplate from 5 to 0.0290625 µg/mL. The medium, growth, and vehicle (DMSO) were controlled. Nystatin was used as a positive control. After adding the *C. albicans* ATCC 10231 inoculum at 2.5 × 10^3^ CFU/mL, the microplate was incubated for 24 h at 35 ± 2 °C. The test was carried out in triplicate. The assay was revealed 2 h after incubation following the addition of 40 µL of 0.01% resazurin.

#### 2.6.3. Agar Diffusion Assay

The test was carried out following CLSI standard M44, with modifications [25]. Initially, the inoculum of *C. albicans* ATCC 10231 was prepared in Sabouraud-Dextrose broth, from colonies grown on Sabouraud-Dextrose agar for 24 h at 35 ± 2 °C and standardized in a spectrophotometer at 530 nm to reach a concentration of approximately 5 × 10^6^ CFU/mL. Using a sterile swab, the inoculum was spread in all directions on Sabouraud-Dextrose agar plates, ensuring that no spaces were left without fungal growth.

Afterward, the sectioned ovules of U-PEO, U-PEO/CUR, U-PEO/MCZ, and U-PEO/CUR/MCZ were placed in the Petri dishes. In one group (n = 3), after placing the materials on the solid medium, the plates were incubated at 35 ± 2 °C. In another group (n = 3), after applying the sectioned ovules to the agar, 100 µL of sterile phosphate buffer pH 7.3 ± 0.2 (PBS; Oxoid Ltd.) was poured over the materials to stimulate the release of the drugs from the formulation. PBS and nystatin were used as controls and deposited at a volume of 50 µL in wells drilled in the agar. After 24 h of incubation at 35 ± 2 °C, the inhibition halos of the tested groups were measured with a pachymeter.

### 2.7. Statistical Analysis

In vitro release and antifungal activity assays were carried out in triplicate and data were expressed as mean ± standard deviation (SD). The triplicate showed no variation in MICs, meaning that no statistical test was necessary. Statistical differences in agar diffusion assay were evaluated by one-way analysis of variance (ANOVA) and Tukey’s multiple comparisons test using GraphPad Prism^®^ version 5.0 software (GraphPad Software, San Diego, CA, USA).

## 3. Results and Discussion

### 3.1. Synthesis of Ureasil–Polyether Hybrid Ovules

The U-PEO vaginal ovules were homogeneous and transparent, with no cracks or fissures after drying. Figure 1 illustrates the visual characteristics of U-PEO both unloaded and loaded with CUR and MCZ ovules.

The visual appearance of the ovules is an important aspect that the visual appearance influences the willingness to try to use of women who need to undergo vaginal treatment to use the ovules. In a study of the literature, round oval ovules were defined as the second preference by women, given the apparent ease of application and the similarity of the shape to other products for vaginal use. Based on these observations, the round ovules made with U-PEO are likely to be well-received by the intended audience [26]. 

Apart from accommodating individual patient preferences, the utilization of cold molding in the production of U-PEO ovules also eliminates errors that may arise from the traditional method of drug production. The latter involves dispersing the drug in a hydrophilic or fatty base, followed by molding and cooling. However, inadequate heating and stirring of the base can lead to drug sedimentation and formulations with non-uniform drug content, which can be avoided when molding takes place without adjusting the temperature [27,28].

Obtaining vaginal ovules by hydrolysis and condensation reactions of poly(ethylene oxide) is an effective approach for loading high quantities of drugs, including thermosensitive drugs, since there is no need to heat the base to melt it, as is required by the conventional method [29]. The U-PEO base can also incorporate lipophilic or hydrophilic compounds without precipitation, as was observed with the incorporation of CUR and micelles containing 3% CUR, and the final material was limpid and homogeneous [22]. In the present study, the matrix proved capable of loading 2.7% of both CUR and MCZ without showing any signs of sedimentation of the drugs.

### 3.2. Physicochemical Characterization of Hybrid Ovules

#### 3.2.1. Thermogravimetry (TGA) 

The thermogravimetric curves of CUR, MCZ, and the unloaded U-PEO ovules incorporated with the drugs are shown in Figure 2, Figure 3, Figure 4 and Figure 5. The TGA parameters are presented in Appendix A.

The TGA curve of U-PEO (Figure 2) showed three stages of thermal decomposition. The first mass loss of approximately 7.53% occurred between 50.20 and 200.70 °C and can be attributed to the evaporation of water and ethanol from the matrix. The second thermal event, between 268.33 and 431.39 °C, with a mass loss of 56.24% is due to the decomposition of the ureasil units. The final thermal event occurred between 446.73 and 545.79 °C with a mass loss of 12.78%. A residue of 23.44% formed by carbonized material was observed, reflecting the incomplete decomposition of the U-PEO hybrid material in the N_2_ flow atmosphere.

In the case of CUR (Figure 3a), stability was maintained up to 200 °C, and there was no mass loss due to dehydration, indicating that the sample was completely dry. The first mass loss of 56.01% occurred between 239.12 and 443.28 °C. The second thermal event, between 440.73 and 900 °C, had a mass loss of 16.03%. According to Chen et al. [30], the decomposition of curcumin takes place in two stages: the first decomposes the substituent groups, and the second decomposes the aromatic rings of the polyphenol structure. CUR left a residue of 26.28% resulting from the formation of carbonized material, which suggests that its thermal decomposition is not complete under a N_2_ atmosphere.

The thermogravimetric analysis of MCZ (Figure 3b) showed decomposition in three stages. The first occurs between 180.42 and 214.16 °C with a loss of 20.49%. The second mass loss of 65.72% occurs between 219.42 and 356.56 °C. The final stage, between 361.40 and 900 °C, results in a 10.01% mass loss. In the first thermal event, C_3_H_3_N_2_ and HCl are released as gases. In the second and final stages, the C_15_H_10_Cl_3_O portion is decomposed. Unlike U-PEO and CUR, MCZ decomposes almost entirely into CO_2_, NO, and CO gases in a N_2_ atmosphere [31].

The TGA curve of the physical binary mixture of CUR and MCZ (Figure 4) showed two thermal events. The first mass loss of 47.58% occurred between 214.40 and 370.75 °C, and the second between 375 and 900 °C with a mass loss of 29.27%. Both refer to the decomposition of the drugs, leaving a final residue of 23.14%. It can be observed that the curve of the binary mixture represents the sum of the curves of the isolated drugs. Therefore, there are no significant changes in the TGA curves and no signs of changes in the mass loss profile of CUR and MCZ undergoing heat stress together.

The U-PEO loaded with CUR and MCZ (Figure 5) showed three thermal events, with the first referring to the loss of water and solvent (40–63 °C) and both the second and third related to the thermal decomposition of the ureasil units of the material and of the incorporated drugs, occurring between 230–273 °C and 435–900 °C, respectively. The curves of the drugs loaded into U-PEO showed a profile similar to the isolated hybrid material, indicating that CUR and MCZ were completely solubilized in the matrix and their decomposition processes occurred simultaneously with those of the hybrid matrix. In addition, the thermal stability of both drugs increased after their incorporation into U-PEO, which was indicated by the increase in the onset temperature of their main thermal decomposition events, when comparing the curves of the isolated drugs (CUR-orange) (MCZ-black) and their physical mixture (CUR + MCZ 1:1-cyan).

#### 3.2.2. Differential Thermal Analysis (DTA)

Figure 6 displays the DTA curves of CUR and MCZ and the pure hybrid precursor combined with the drugs. The DTA parameters are shown in Appendix A.

The hybrid precursor PEO is amorphous and has no defined melting temperature. Also, no significant peak events were observed in its curve [32]. Analyzing the physical mixture of the hybrid precursor with CUR and MCZ (green curve), it was observed that the material’s amorphous characteristic was predominant. This may indicate that the dissolving of the drugs in the liquid hybrid precursor has decreased their crystallinity to undetectable levels or that the drugs have passed into an amorphous state.

CUR exhibits an endothermic peak at 186.41 °C with an enthalpy of ΔH = −77.11 J/g. The peak is not associated with a mass loss in the TGA curve and indicates the drug’s melting temperature. The second event is exothermic and occurs at 381.34 °C with ΔH = 23.79 J/g. This event is related to the loss of mass from CUR’s main thermogravimetric event and the loss of its substituent groups due to thermal stress [33].

MCZ curve showed an endothermic and an exothermic peak. The first one occurs at 185.29 °C and represents the melting point of the drug, with an ΔH equal to −42.81 J/g. The second peak, at 208.25 °C (ΔH = 252.56 J/g), is exothermic and can be attributed to partial recrystallization resulting from a second-order transition to the molten state from an amorphous state, based on what was observed in the differential scanning calorimetry (DSC) analysis of MCZ [34].

The thermogram of the binary mixture of CUR and MCZ 1:1 showed a single endothermic peak at 161.79 °C (ΔH = −58.99 J/g), suggestive of the melting point of the drugs. The reduction in temperature of the melting peak and the suppression of the exothermic peaks previously presented in the CUR and MCZ curves alone may indicate physical incompatibility between them due to the formation of a eutectic mixture. This interaction may result from intermolecular hydrogen bonding between the phenolic hydroxyls and carbonyls present in CUR, the nitrogen of the imidazole ring, and the oxygen of the ether in MCZ.

In the curve of the physical mixture of CUR and the hybrid precursor, it was observed that PEO anticipated the endothermic peak of CUR (T_peak_ = 111.67 °C) and reduced its intensity (ΔH = −4.66 J/g), indicating a physical interaction between the hybrid precursor and the active compound. The CUR structure has chemical groups such as phenolic hydroxyls and diacetone capable of forming intramolecular hydrogen bonds with the urea and oxyethylene units in the PEO structure. The reduction in the melting point of CUR in this case has no clinical or technological significance, since thermal incompatibility occurs at temperatures above 100 °C, not affecting the administration of the formulation nor the process of mixing CUR with the hybrid precursor to produce ovules.

In the MCZ mixture with the hybrid precursor, the exothermic peak of the drug was subtly anticipated to T_peak_ = 194.52 °C, while the endothermic peak was not detected. This behavior suggests that the drug may have changed from a crystalline to an amorphous state, resulting in the recording of only the exotherm of crystallization [35]. 

Despite the observed melting point depression, as shown by the DTA curves of the binary mixture of CUR and MCZ, and CUR and PEO, as well as the suppression of the MCZ endothermic peak in the physical mixture with the hybrid precursor, these changes were a result of physical interactions under heat stress conditions. However, they do not affect the production of vaginal ovules from the hybrid precursor, nor the co-incorporation of CUR and MCZ in the final U-PEO matrix.

#### 3.2.3. Fourier Transform Infrared Spectroscopy (FTIR)

The FTIR spectra of U-PEO (Figure 7) showed a broad band at 3370 cm^−1^, related to the stretching of the N-H bonds of the urea groups. The bonds of these groups can also be visualized by bands in the amide I and II regions at 1635 cm^−1^ and 1560 cm^−1^, respectively. The absorptions at 2918 and 2878 cm^−1^ correspond to the asymmetric and symmetric C-H stretching region. The most intense absorption peak (1080 cm^−1^) can be attributed to the stretching of C-O bonds. 

The CUR scan (Figure 8) showed characteristic peaks of O-H stretching of phenolic hydroxyls at 3508 and 3380 cm^−1^, stretching of C=C bonds conjugated to C=O at 1625 cm^−1^, referring to diacetone groups, aromatic C=C stretching at 1600 cm^−1^, C-H bending of methyl groups at 1427 cm^−1^, and C-O bond stretching at 1273 cm^−1^ [36,37].

In MCZ spectra (Figure 8), absorption peak characteristics of C-H stretching were observed between 3182–3107 cm^−1^, which can be attributed to the aromatic rings or the imidazole. Pairwise absorptions at 1586 and 1545 cm^−1^ refer to aromatic and imidazole C=C stretching, while the peaks at 1330 cm^−1^ and 1082 cm^−1^ represent C=N and C-O stretching vibrations, respectively [38]. 

Figure 8 displays the spectra of the binary mixture of CUR and MCZ in a 1:1 ratio. It is possible to detect the presence of the original bands of both drugs. The peaks referring to the phenolic O-H stretch at 3508 and 3380 cm^−1^ of CUR, as well as the stretching bands of C=C bonds conjugated to C=O of the diacetone groups at 1625 cm^−1^ were observed without shifting. The peak referring to the aromatic C=C stretch of CUR at 1600 cm^−1^ could not be detected due to interference from the absorbances of MCZ in the same region. The C-O stretching peak at 1273 cm^−1^ was subtly shifted to 1280 cm^−1^. This may also result from overlapping bands with the MCZ spectrum, given that no azole bands were shifted to indicate possible interaction with this CUR chemical group.

Analyzing the scan of the physical mixture with CUR, all the MCZ bands remained unchanged and exhibited no shift. The C-H stretching peaks at 3182 and 3107 cm^−1^ of the imidazole ring were observed at a lower intensity. Additionally, the C=C stretching bands of the rings were also detected. No changes were observed in the C=N and C-O stretching peaks of the drug. The decrease in the intensity of the original CUR and MCZ bands indicates that the mass of the compounds in the physical mixture is reduced when compared to their isolated spectra.

No new bands were detected nor were the original bands were replaced, indicating that there was no chemical interaction or formation of new chemical groups between CUR and MCZ in the physical mixture. Based on the results of the differential thermal analysis, it can be confirmed that the incompatibility issue between CUR and MCZ only arises under thermal stress. Therefore, it should be noted that this does not affect the formulation process when these drugs are combined without heating.

In the FTIR spectrum of U-PEO ovules incorporated with CUR and/or MCZ (Figure 9), it was possible to observe the overlapping of the chemical groups of the drugs with that of the chain of the hybrid material, predominantly in all the scans. The band at 3370 cm^−1^ referring to the N-H stretching of the urea groups was split into two lower-intensity components. Moreover, the C-H bond stretching region showed an increase in the intensity of the peak at 2278 cm^−1^, potentially indicating a modification of the nanostructure of the hybrid material to accommodate the drugs [39].

In addition, the increase in intensity of the bands in the amide I and II regions (1635 and 1560 cm^−1^) also suggests changes in the structure of U-PEO after incorporation of the actives. These bands are related to urea groups and urea–urea interactions within the matrix. The amide I band represents a multi-component complex such as stretching and deformation of C-N bonds and stretching of C=O groups that are not involved in urea–urea intermolecular bonds. The amide II band reflects the bending vibration in the N-H plane and is related to the conformation of the chain and intermolecular hydrogen bonds [40]. In this sense, the decrease in intensity of amide I after the incorporation of CUR and MCZ into U-PEO may indicate a decrease in free urea carbonyl groups due to interaction with the drugs through hydrogen bonds. In contrast, the increase in amide II intensity suggests a modification in the urea–urea intermolecular bonds between the U-PEO chains. This result agrees with the change in the C-H bond stretching band, indicating a conformational shift after drug incorporation.

After CUR and MCZ were incorporated, the characteristic C-O stretching peak of the oxyethylene groups of U-PEO became stronger, shifting from 1080 cm^−1^ to 1088 cm^−1^. The appearance of a shoulder in this band, located at 1030 cm^−1^, was also observed. These changes could have occurred due to the overlapping of the CUR and MCZ bands that also have C-O stretching groups and bands from their fingerprint region, which appear in this wavenumber range.

### 3.3. In Vitro Drug Release Assay

Figure 10 shows the release curves of CUR and MCZ. The graph indicates that 42% of CUR was released in 24 h, reaching 61% within 48 h. After 120 h the curve reached a plateau with 81% of the polyphenol released. For MCZ, a similar pattern was observed, with 57% of the drug released in the first 24 h and 70% within 48 h. The drug continued to release, reaching 75% at 72 h and 95% at the end of 144 h. Thus, the release profile of CUR and MCZ from U-PEO proved to be prolonged.

An important feature of the release profile of CUR and MCZ from U-PEO is that approximately 40% of the drugs (representing around 54 mg) are released in the first 24 h. The release follows a prolonged profile until the end of six days. The initial concentration released is important for promoting initial antifungal activity and relieving symptoms, while the percentage released later may maintain activity for controlling and curing the infection.

In addition, the prolonged release profile of the antifungal agents in the formulation can reduce the frequency of use compared to commercially available formulations, which require daily administration for an average of three to seven days [41]. This could significantly contribute to reducing the risk of therapeutic failure associated with the interruption of treatment before the mycological cure, and could increase women’s compliance with therapy, resulting in improved patient outcomes.

The CUR and MCZ release data were fitted to four mathematical models to study the transport mechanisms that govern the release of the drugs. Based on the highest regression coefficient values (R^2^), the models to which the release curves fitted best were determined [39].

The MCZ release fitted the Korsmeyer-Peppas model best. This semi-empirical model was developed to describe the release of drugs from polymeric matrices and is governed by the equation *Mt = kt^n^*, where *Mt* corresponds to the total amount released, *k* is the model constant, and *n* is the exponent related to the solute transport mechanism [42]. The value of n calculated for the MCZ curve was 0.3303, indicating that the drug is released from the matrix by a pseudo-Fickian diffusion mechanism.

On the other hand, the CUR release data showed the highest R^2^ for the Weibull model. Although this is not a well-founded model for describing drug release kinetics, the release data can be adjusted according to the equation *Mt* = 1 − *exp* (−*at^b^*), where *Mt* is the total amount released, *a* and *t* are constants, and the exponent *b* is the shape parameter related to the type of curve. In this expression, *b* has a linear relationship with the exponent *n* of the Korsmeyer-Peppas model and its value can indicate the type of drug transported from the matrix. In the CUR curve, the value of *b* was equal to 0.5515 and the drug was released by Fickian diffusion (*b* ≤ 0.75) [43].

### 3.4. Antifungal Activity of CUR and MCZ

#### 3.4.1. Determination of the Minimum Inhibitory Concentration (MIC)

The antifungal activity of CUR and MCZ was evaluated using the microdilution method with 0.01% resazurin. The wells that did not change the resazurin color from purple to pink and had the lowest concentration of CUR and MCZ were considered the minimum inhibitory concentrations. 

The minimum concentrations of CUR and MCZ capable of inhibiting the growth of *C. albicans* ATCC 10231 were 256 µg/mL and 2.5 µg/mL. The negative, positive, vehicle, and growth controls were adequate.

These findings corroborate data in the available literature. The MIC of CUR by the broth microdilution method was also equal to 256 µg/mL against *C. albicans* ATCC 10231 in the study conducted by Xue et al. [44]. In another study, MCZ’s MIC50 and MIC80, defined as the lowest concentrations to inhibit growth by 50% to 80%, were equal to 2 µg/mL and 4 µg/mL, in this order, against the same reference strain of *C. albicans* [45].

#### 3.4.2. Checkerboard Test for Synergism Evaluation

The interaction between CUR and MCZ against *C. albicans* ATCC 10231 was evaluated using the checkerboard microdilution method. The test results are shown in Table 1.

The MIC found for CUR alone, equal to 256 µg/mL, decreased eight-fold when combined with MCZ, resulting in 32 µg/mL. In turn, the MIC of MCZ showed a four-fold reduction, reaching a value of 0.625 µg/mL when the azole antifungal was combined with the polyphenol. 

The fractional inhibitory concentration index (FICI), obtained from the sum of the FICs of CUR and MCZ, was equal to 0.375 and confirmed the synergistic effect between the drugs, as indicated when this value is less than or equal to 0.5. 

CUR’s antifungal activity is mainly due to the generation of reactive oxygen species (ROS) which induce apoptosis in *C. albicans*. MCZ acts primarily by inhibiting ergosterol biosynthesis, which results in the accumulation of toxic levels of methylated sterols. This azole is also the only agent in its pharmacological class capable of generating a fungicidal effect through the accumulation of ROS [46,47].

The combination of CUR and MCZ shows activity by potentiating the effects of the generation of reactive oxygen species (ROS) in *C. albicans* cells presented by both drugs. This occurs at much lower concentrations than when the polyphenol is used alone. The combined therapy also results in early apoptosis of viable forms of the fungus. By acting on various signaling pathways, this therapy may help prevent the development of resistance to azole drugs by *Candida* [47]. 

In parallel with potentiating the antifungal effect of MCZ, CUR can inhibit efflux pumps responsible for azole resistance such as CaCdr1p and CaCdr2p, without serving as a target. Thus, this phytoconstituent is capable of sensitizing fungal cells to the imidazole, which in turn prevents the microorganism from eliminating it and thereby avoiding therapy failure caused by resistant fungal strains [48].

The co-administration of antifungal agents is effective in increasing the pharmacological effect but can cause adverse effects, such as vulvovaginal burning sensation, reported after the topical administration of CUR, as well as irritation, burning, and itching with the application of MCZ. However, when the effect of the combined drugs is synergistic, the doses can be reduced and, consequently, the adverse effects resulting from both drugs can be minimized [49,50,51]. 

Therefore, combination therapy is a promising approach that can minimize toxicity, treat multiple fungal infections, effectively control the infection, increase the possibility of curing it, as well as delay or reduce the development of fungal resistance. This approach has the potential to be a valuable tool in the development of novel treatment options for fungal infections [51].

### 3.5. Antifungal Activity of U-PEO Ovules

To evaluate the antifungal activity of pure U-PEO and its possible interaction with the effect of CUR or MCZ, a broth microdilution assay was conducted. After revealing with resazurin, the triplicate wells of the isolated U-PEO group showed a pink color, indicating that none of the hybrid material concentrations were able to inhibit fungal growth. Visual analysis of the wells also confirmed the growth of colonies.

In contrast, the presence of crushed hybrid material did not affect the minimum inhibitory concentrations of CUR or MCZ in all replicates, which remained at 256 µg/mL and 2.5 µg/mL, respectively, in the U-PEO/CUR and U-PEO/MCZ groups. All controls were deemed suitable at the conclusion of the experiment.

There is limited literature available on antimicrobial tests conducted using ureasil–polyether materials against *C. albicans*. However, a study on films made of ureasil/PPO-PEO-PPO triblock copolymer incorporated with human intragenic antimicrobial peptides showed that the blank hybrid material did not exhibit any antimicrobial activity against *Escherichia coli* in a liquid culture medium test [52].

The anti-*Candida* activity of U-PEO, U-PEO/CUR, U-PEO/MCZ, and U-PEO/CUR/MCZ ovules was also evaluated in a solid culture medium by agar diffusion. This experiment was divided into two groups: one with the deposition of 100 µL of PBS under the ovule fragments, and one without it. The zones of inhibition were measured after 24 h of incubation, and the results are shown in Figure 11 and Appendix A.

In both groups, the isolated hybrid material showed no inhibition halo, corroborating the microdilution results on the lack of U-PEO activity. In the absence of PBS, the U-PEO/CUR halo was 2.25 mm ± 0.86 mm and showed no statistical difference from U-PEO. This indicates that, due to its hydrophobic nature, CUR did not diffuse effectively into the hydrophilic agar medium, preferring to be retained in the hybrid matrix. In turn, U-PEO/MCZ and U-PEO/CUR/MCZ showed inhibition zones of 22.5 mm ± 2.88 mm and 18.5 mm ± 0.57 mm, respectively, with no statistical difference between them. All groups without PBS showed significantly different results from the positive control nystatin, which inhibited growth to a diameter of 28.5 mm ± 1.73 mm.

After applying PBS to the ovules, the diffusion of the drugs into the solid medium seems to have been improved and the zones of inhibition were greater. The U-PEO/CUR halo was significantly larger in this group, measuring 15 mm ± 5.77 mm in diameter. The inhibition of *C. albicans* by U-PEO/MCZ and U-PEO/CUR/MCZ was measured at 29.5 mm ± 0.57 mm and 24.5 mm ± 2.88 mm, in that order, with no statistical difference between them or with nystatin. The PBS control did not inhibit fungal growth.

According to CLSI standard M44, antifungal susceptibility can be interpreted by the size of the zone of inhibition. The sensitivity of a strain is determined by halos measuring 20 mm or more. Zones of inhibition between 15 and 19 mm may suggest dose-dependent or intermediate susceptibility, while inhibition less than or equal to 14 mm in diameter points to the resistance of the tested strain to the antimicrobial agent [25].

In our study, the inhibition zones of U-PEO/CUR with and without PBS can be classified as intermediate and resistant susceptibility, respectively. U-PEO/MCZ ovules showed zones of inhibition compatible with sensitivity in the groups with and without PBS. In turn, U-PEO/CUR/MCZ exhibited a halo of susceptibility only in the group that received the buffer. The *C. albicans* ATCC 10231 strain also showed sensitivity to the positive control nystatin. The lower diffusion rate of CUR from U-PEO to agar resulted in lower susceptibility compared to the other ovules, but U-PEO containing MCZ and CUR/MCZ exhibited activity comparable to that of the commercial drug nystatin.

Along with being as effective at inhibiting fungal growth as nystatin (an antifungal also used in the treatment of VVC), MCZ is clinically superior when it comes to cure rate, fewer adverse effects, shorter treatment time, and infection recurrence rate. The clinical relevance of MCZ has also been shown to be better than other drugs of its class, such as clotrimazole, which has a lower cure rate and a higher risk of relapse [53,54]. The therapeutic efficacy of MCZ can also be enhanced by synergism with CUR, which acts by complementary mechanisms such as inhibiting hyphal development, altering membrane permeability, and inhibiting biofilms [55]. Therefore, CUR and MCZ combined in a modified-release formulation based on U-PEO can contribute positively to improving the clinical response to the treatment of VVC. 

## 4. Conclusions

Our study presents a new vaginal ovule combining curcumin and miconazole for the treatment of vulvovaginal candidiasis caused by *C. albicans*. This is the first time this synergistic combination has been used in a vaginal ovule. The ovules were produced using a hybrid ureasil–polyether base through cold molding, resulting in a uniform drug content. Curcumin and miconazole combined exhibited antifungal activity at concentrations eight and four times lower than when used separately. This suggests that lower doses of the drugs could be used, which could reduce adverse effects. The U-PEO matrix offers prolonged drug release and may allow less frequent drug administration, potentially increasing compliance with treatment. However, studies in animal models are required to confirm these results in conditions that mimic the vaginal environment in vivo and to compare the efficacy of the new formulation with commercial treatments, therefore allowing the research to progress to pre-clinical and clinical trials.

## Figures and Tables

**Figure 1 pharmaceutics-16-00312-f001:**
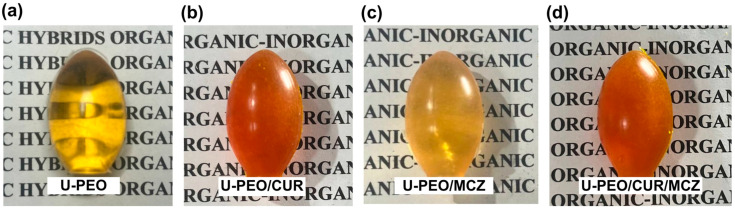
U-PEO vaginal ovules. (**a**) U-PEO, (**b**) U-PEO-CUR, (**c**) U-PEO-MCZ, (**d**) U-PEO CUR-MCZ.

**Figure 2 pharmaceutics-16-00312-f002:**
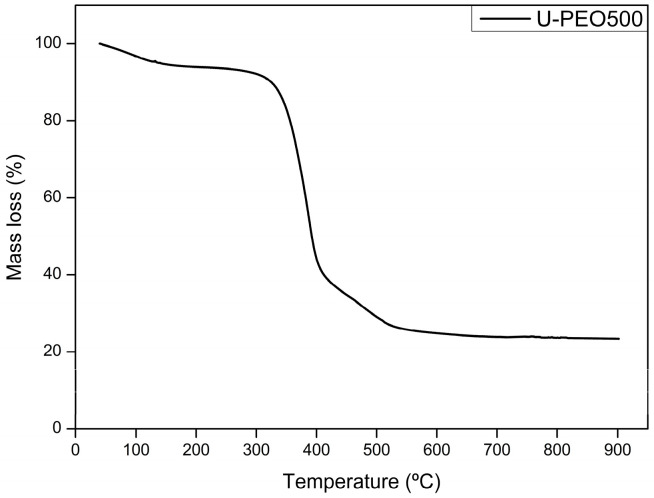
Thermogravimetric curve of U-PEO hybrid material.

**Figure 3 pharmaceutics-16-00312-f003:**
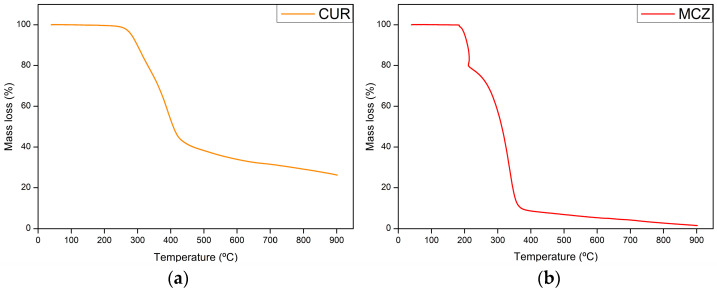
Thermogravimetric curves of (**a**) CUR and (**b**) MCZ.

**Figure 4 pharmaceutics-16-00312-f004:**
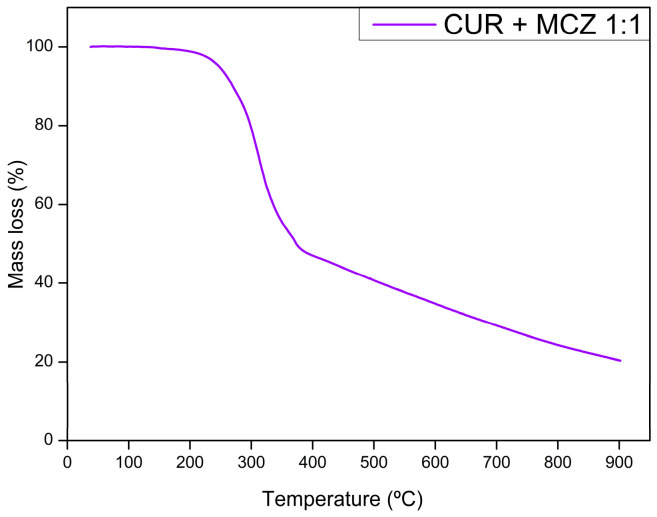
Thermogravimetric curves of the binary mixture CUR and MCZ 1:1.

**Figure 5 pharmaceutics-16-00312-f005:**
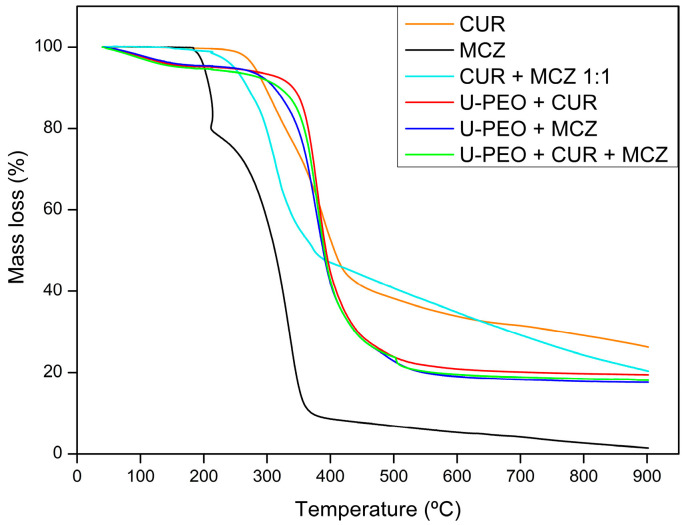
Thermogravimetric curves of CUR (orange), MCZ (black), binary mixture of CUR and MCZ (cyan), CUR loaded in U-PEO (red), MCZ loaded in U-PEO (blue), and U-PEO loaded with CUR and MCZ (green).

**Figure 6 pharmaceutics-16-00312-f006:**
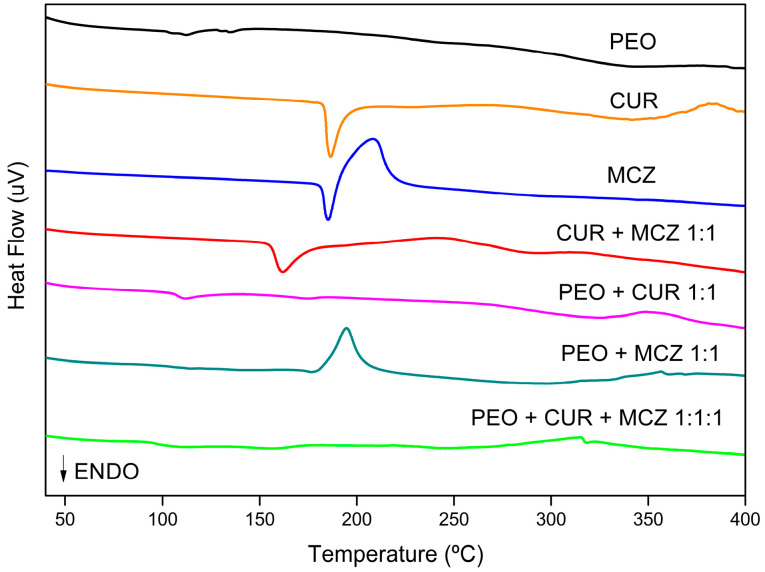
DTA curves of hybrid precursor PEO (black), CUR (orange), MCZ (blue), the binary mixture CUR + MCZ 1:1 (red), binary mixtures of PEO and CUR (pink), PEO and MCZ (dark cyan) and PEO with CUR and MCZ (green).

**Figure 7 pharmaceutics-16-00312-f007:**
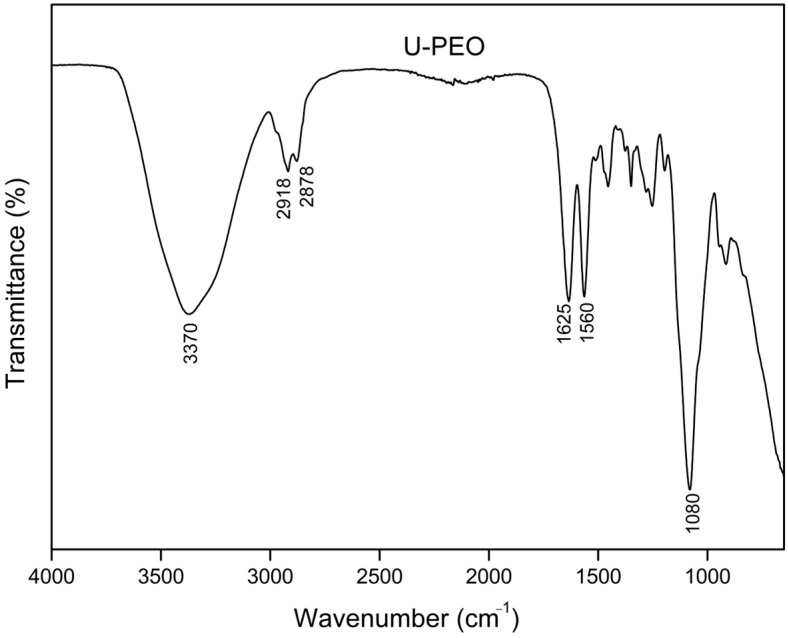
FTIR spectra of U-PEO.

**Figure 8 pharmaceutics-16-00312-f008:**
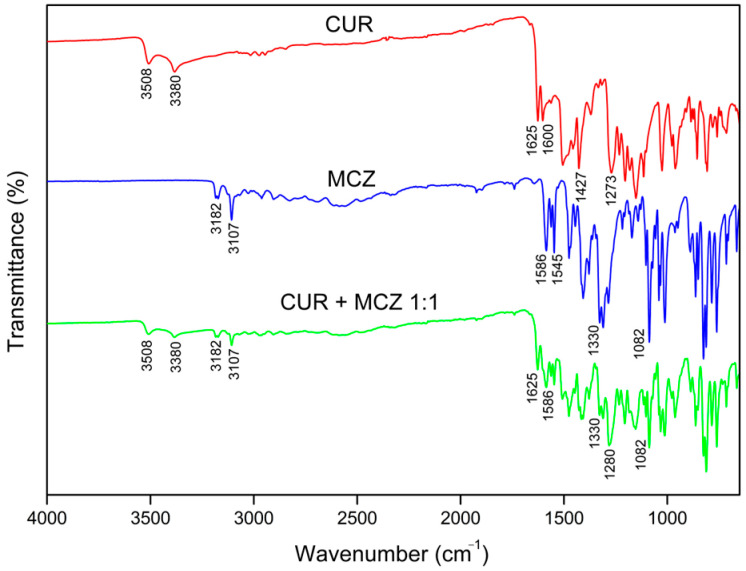
FTIR spectra of CUR (red), MCZ (blue), and binary mixture CUR + MCZ 1:1 (green).

**Figure 9 pharmaceutics-16-00312-f009:**
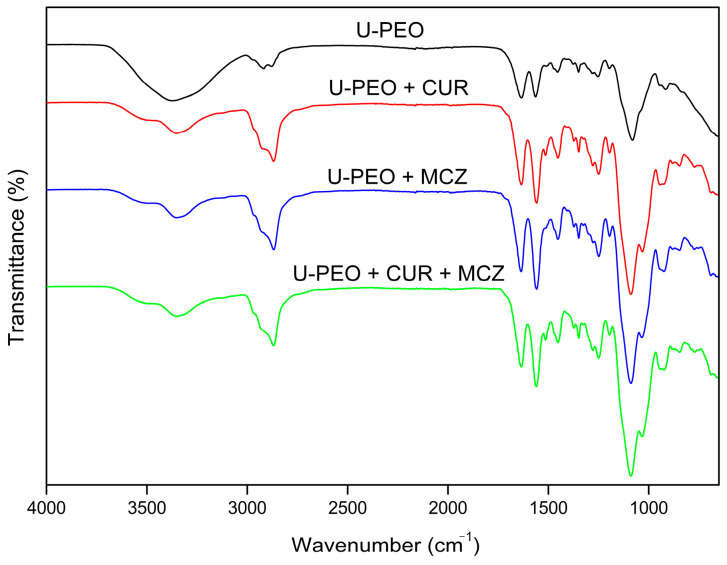
FTIR spectra of unloaded U-PEO (black) and U-PEO loaded with CUR (red), MCZ (blue), and CUR + MCZ (green).

**Figure 10 pharmaceutics-16-00312-f010:**
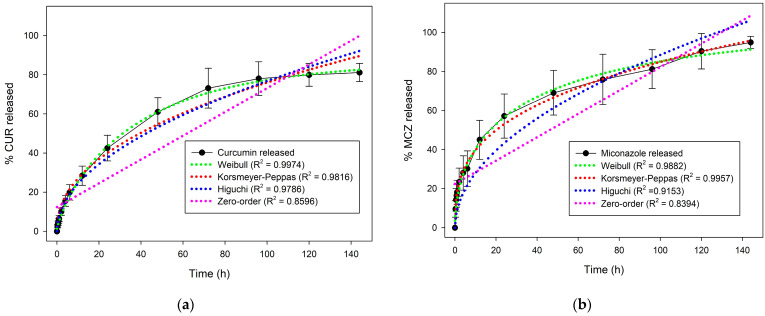
In vitro release curves of CUR and MCZ: (**a**) U-PEO + CUR, fitting CUR curve to different mathematical models: Weibull (green), Korsmeyer-Peppas (red), Higuchi (blue), and zero-order (pink); (**b**) U-PEO + MCZ, fitting MCZ curve to different mathematical models: Weibull (green), Korsmeyer-Peppas (red), Higuchi (blue), and zero-order (pink). Data are presented as mean ± SD (n = 3).

**Figure 11 pharmaceutics-16-00312-f011:**
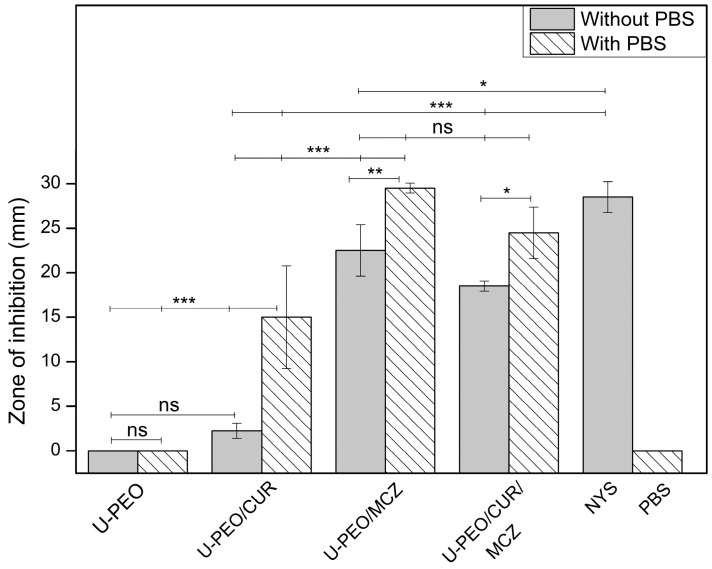
Zones of inhibition of U-PEO, U-PEO/CUR, U-PEO/MCZ, and U-PEO/CUR/MCZ against *C. albicans* ATCC 10231 in agar diffusion assay. NYS (nystatin) and PBS (Phosphate buffer) represent control groups. Data are expressed as mean ± SD (n = 3). * *p* < 0.05; ** *p* < 0.01; *** *p* < 0.001; ns: no significant difference.

**Table 1 pharmaceutics-16-00312-t001:** Evaluation of the synergistic effect between CUR and MCZ against *C. albicans*.

Microorganism	Drug Tested	MIC (µg/mL)		FICI	Result
*C. albicans* ATCC 10231	CUR MCZ	Alone	Combination	0.375	Synergism
		256 2.5	32 0.625		

## Data Availability

The data presented in this study are available in this article and Appendix A.

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
