# Peer review of "The Efficacy of Hybrid Vaginal Ovules for Co-Delivery of Curcumin and Miconazole against Candida albicans"

_pharmaceutics, 2024, doi:10.3390/pharmaceutics16030312_

Round 1
Reviewer 1 Report
Comments and Suggestions for Authors
In this study, it was developed ureasil-poly(ethylene oxide) (U-PEO) vaginal ovules loaded with CUR and MCZ as a new treatment for vulvovaginal candidiasis caused by albicans. However, the study has mistakes that make it unacceptable.
1. Why the antifungal effect of U-PEO-MCZ, U-PEO-CUR has not been evaluated? It is possible that U-PEO itself has an antifungal effect, so it seems necessary to evaluate the antifungal effect of U-PEO-MCZ, U-PEO-CUR and U-PEO.
2. Antifungal effect test should be repeated three times so that its standard deviation is calculated and the results are reported in terms of statistical significance.
3. A section for statistical methods to evaluate the results is not considered.
4. The contents of Table 1 can be simply written in the text and there is no need for a table.
5. Figure 4 is not mentioned in the text.
6. Manuscript should be updated with new resources from 2022-23.
7. In evaluating the antifungal effect, a control for dimethyl sulfoxide (DMSO) as a vehicle has not been considered.
Moderate editing of English language required
Author Response
We would like to thank the reviewers for their valuable comments on our work. All issues raised in the review process are addressed in the Revised Manuscript. Corrections suggested by Reviewer 1 are highlighted in gray, those suggested by Reviewer 2 are highlighted in cyan, and those recommended by Reviewer 3 are highlighted in yellow. Other modifications made by the authors are highlighted in green in the text. The revised paper brings new experiments to evaluate the antifungal activity of U-PEO ovules, and more details to make it more didactic and clear for the reader.
In this study, it was developed ureasil-poly(ethylene oxide) (U-PEO) vaginal ovules loaded with CUR and MCZ as a new treatment for vulvovaginal candidiasis caused by albicans. However, the study has mistakes that make it unacceptable.
- Why the antifungal effect of U-PEO-MCZ, U-PEO-CUR has not been evaluated? It is possible that U-PEO itself has an antifungal effect, so it seems necessary to evaluate the antifungal effect of U-PEO-MCZ, U-PEO-CUR and U-PEO.
R: Thank you for your attention. Your point is correct. To evaluate the antifungal activity of the hybrid material U-PEO pure and with CUR and MCZ, we carried out two methods: microdilution and agar diffusion. For the first, we triturated the U-PEO ovules and solubilized them in 10% DMSO, then carried out the test on the microplate and checked the activity of the pure material and when physically mixed with the drugs. For the second method, we sectioned the ovules into smaller pieces. We deposited them on Sabouraud-dextrose agar after sowing the inoculum, evaluating the inhibition zones at the end of the incubation period. In both tests, we observed that U-PEO could not inhibit fungal growth and therefore has no activity against C. albicans or even synergism with CUR or MCZ. The presence of U-PEO also did not alter the MICs of the drugs in microdilution. The new experiments are described in section 2.6 of the methods and 3.5 in the results and discussion. The new sections are highlighted in gray in the text.
- Antifungal effect test should be repeated three times so that its standard deviation is calculated and the results are reported in terms of statistical significance.
R: We appreciate your attention. The antifungal activity tests were performed in triplicate on a single microplate (n = 3), but the minimum inhibitory concentration (MIC) results remained consistent. The agar diffusion tests were also conducted in triplicate, and statistical tests were applied. The results of these tests are presented as mean ± standard deviation (SD) in the paper.
- A section for statistical methods to evaluate the results is not considered.
R: Thank you for your observation. We have added the statistical analysis in a new section (2.7), highlighted in gray in the manuscript.
- The contents of Table 1 can be simply written in the text and there is no need for a table.
R: Thank you for your comment. Table 1 has been removed from the manuscript and the results are written in the text as recommended.
- Figure 4 is not mentioned in the text.
R: Thank you for your attention. We have added the mention of figure 4 in the text (line 307) and highlighted the passage in gray.
- Manuscript should be updated with new resources from 2022-23.
R: We have updated some of the references in the manuscript so that at least half of the list consists of current sources published between 2020 and 2023 at the earliest. However, some crucial references for our study were published some time ago and we have to maintain them in the document. The updated references are highlighted in gray in the text and in the list.
- In evaluating the antifungal effect, a control for dimethyl sulfoxide (DMSO) as a vehicle has not been considered.
R: Thank you for your observation. We noticed that we had mentioned the controls in the results and discussion section, but not in the methods, and we apologize for the mistake. Vehicle control was considered in the tests. We have corrected the writing of the section to include the controls that were used in the antifungal activity experiments.
Reviewer 2 Report
Comments and Suggestions for Authors
The paper is well written. In my opinion only description of the method in section 2.5.2. (lines 167-170) could be supplemented with an information about the time of incubation with resazurin, after which Authors noticed the color change.
Author Response
We would like to thank the reviewers for their valuable comments on our work. All issues raised in the review process are addressed in the Revised Manuscript. Corrections suggested by Reviewer 1 are highlighted in gray, those suggested by Reviewer 2 are highlighted in cyan, and those recommended by Reviewer 3 are highlighted in yellow. Other modifications made by the authors are highlighted in green in the text. The revised paper brings new experiments to evaluate the antifungal activity of U-PEO ovules, and more details to make it more didactic and clear for the reader.
- The paper is well written. In my opinion only description of the method in section 2.5.2. (lines 167-170) could be supplemented with an information about the time of incubation with resazurin, after which Authors noticed the color change.
R: Thank you for your comment. After adding the resazurin, we incubated the microplates for two hours and then analyzed the color change. we have modified the text in section 2.5.2 to add this incubation temperature and highlighted the change in cyan.
Reviewer 3 Report
Comments and Suggestions for Authors
The authors of the manuscript “Efficacy of hybrid vaginal ovules for co-delivery of curcumin and miconazole against Candida albicans” present an interesting work about the use of develop ureasil-polyether (U-PEO) vaginal ovules loaded with curcumin and miconazole for the treatment of vulvovaginal candidiasis. Physicochemical analysis and in vitro antifungal assays such as MIC e FICI were carried out. The in vitro results are promising.
The manuscript is clearly written, and the results are of interest but, it contains some little flawn to be improved before acceptance.
Points that need to be addressed:
Section 3 - Results: Please, probably it is better to rename the section. Results and Discussion section maybe better.
Section 3.1 and 3.2: Please, delete the txt inside the figures (1-9). Text increase confusion; the legend of the figures are ok. Wavenumbers and other symbols are admitted.
Section 3.2.1: Please, may authors explain how did they choose the limits of the thermal events? Especially between the second and third one in figure 2 (slope changes?). May you put this rule on materials and methods section?
Section 3.4: Please, may authors add a sentence (or little bit more) with a comparison between MCZ and another used antifungal compound? Not a new experiment, just data from bibliography for better focusing the importance of CUR, MCZ and their synergic mix.
Author Response
We would like to thank the reviewers for their valuable comments on our work. All issues raised in the review process are addressed in the Revised Manuscript. Corrections suggested by Reviewer 1 are highlighted in gray, those suggested by Reviewer 2 are highlighted in cyan, and those recommended by Reviewer 3 are highlighted in yellow. Other modifications made by the authors are highlighted in green in the text. The revised paper brings new experiments to evaluate the antifungal activity of U-PEO ovules, and more details to make it more didactic and clear for the reader.
The manuscript is clearly written, and the results are of interest but, it contains some little flawn to be improved before acceptance.
Points that need to be addressed:
Section 3 - Results: Please, probably it is better to rename the section. Results and Discussion section maybe better.
R: Thank you for your observation. We promptly changed the name of the section to "Results and discussion".
Section 3.1 and 3.2: Please, delete the txt inside the figures (1-9). Text increase confusion; the legend of the figures are ok. Wavenumbers and other symbols are admitted.
R: If the reviewer agrees with our reasoning, we would rather keep the texts in the figures mentioned, since they serve as a guide to the curves they represent in the physicochemical characterization analyses, and can facilitate the readers' understanding of the data.
Section 3.2.1: Please, may authors explain how did they choose the limits of the thermal events? Especially between the second and third one in figure 2 (slope changes?). May you put this rule on materials and methods section?
R: Thank you for your attention. The limits of thermal events can be determined by analyzing the changes in the slopes of the curves. Still, a more appropriate way to determine this is to plot the derivative curve (DTG), in which the curvatures of the TGA graph are replaced by peaks, making it easier to determine the beginning and end of a thermal event. We have added a description of how to determine thermal events in the methodology and highlighted the new section in yellow.
Section 3.4: Please, may authors add a sentence (or little bit more) with a comparison between MCZ and another used antifungal compound? Not a new experiment, just data from bibliography for better focusing the importance of CUR, MCZ and their synergic mix.
R: Since we have carried out further experiments to evaluate the antifungal activity of U-PEO ovules, we have taken the opportunity to add the paragraph requested in the discussion of these new tests. The text is shown in the new section 3.5 and highlighted in yellow and written below.
New text: “Besides being as effective at inhibiting fungal growth as nystatin, which is also used in the treatment of VVC, MCZ is clinically superior when it comes to cure rate, fewer adverse effects, shorter treatment time, and infection recurrence rate. The clinical relevance of MCZ has also been shown to be better than other drugs of its class, such as clotrimazole, which has a lower cure rate and a higher rate of relapse [53,54]. The therapeutic efficacy of MCZ can also be enhanced by synergism with CUR, which acts by complementary mechanisms, such as inhibiting hyphal development, altering membrane permeability, and inhibiting biofilms [55]. Therefore, CUR and MCZ combined in a modified-release formulation based on U-PEO can contribute positively to improving the clinical response to the treatment of VVC.”
Round 2
Reviewer 1 Report
Comments and Suggestions for Authors Corrections have been made and the manuscript is acceptable.